# Atomic resolution electron microscopy in a magnetic field free environment

N. Shibata [1,2], Y. Kohno[3], A. Nakamura[3], S. Morishita [3], T. Seki[1], A. Kumamoto [1], H. Sawada[3], T. Matsumoto[1], S.D. Findlay [4] & Y. Ikuhara [1,2]

Atomic-resolution electron microscopes utilize high-power magnetic lenses to produce magnified images of the atomic details of matter. Doing so involves placing samples inside the magnetic objective lens, where magnetic fields of up to a few tesla are always exerted. This can largely alter, or even destroy, the magnetic and physical structures of interest. Here, we describe a newly developed magnetic objective lens system that realizes a magnetic field free environment at the sample position. Combined with a higher-order aberration corrector, we achieve direct, atom-resolved imaging with sub-Å spatial resolution with a residual magnetic field of less than 0.2 mT at the sample position. This capability enables direct atom-resolved imaging of magnetic materials such as silicon steels. Removing the need to subject samples to high magnetic field environments enables a new stage in atomic resolution electron microscopy that realizes direct, atomic-level observation of samples without unwanted high magnetic field effects.

[1] Institute of Engineering Innovation, The University of Tokyo, Bunkyo, Tokyo 113-8656, Japan. [2] Nanostructures Research Laboratory, Japan Fine Ceramic Center, Atsuta, Nagoya 456-8587, Japan. [3] JEOL Ltd., Akishima, Tokyo 196-8558, Japan. [4] School of Physics and Astronomy, Monash University, Melbourne, VIC 3800, Australia. Correspondence and requests for materials should be addressed to N.S. (email: shibata@sigma.t.u-tokyo.ac.jp)

In the 88 years since the seminal invention of the transmission electron microscope (TEM) by Ruska and Knoll in 1931[1], researchers have always pursued better and better spatial resolution. As electrons are charged particles, electromagnetic fields have been utilized as lenses for electrons following the discovery of Busch in 1926[2]. Because of their stability and controllability, high-power magnetic lenses have been considered the best lens systems for modern TEMs. Historically, the spatial resolution of TEMs has been strongly limited not by the wavelength of the incident electrons but rather by the performance of the magnetic objective lens: it forms the primary image and diffraction pattern, which will be magnified by all the other lenses, and so the quality of the final image is severely affected by its lens aberrations. To achieve better spatial resolution, the design of magnetic objective lenses with smaller lens aberration coefficients (such as for spherical aberration, $C_s$, and chromatic aberration, $C_c$) has been at the heart of modern TEM developments. The 1995 development by Haider et al.[3] of a practical spherical aberration correcting lens systems produced breakthrough improvements in spatial resolution[4]. Since then, aberration correcting lens systems for scanning TEM (STEM)[5] have achieved sub-Å spatial resolution in real-space[6], with sub-0.5 Å resolution having been recently reported with 300 kV accelerating voltage[7]. Nowadays, individual single atoms can be routinely imaged by commercially available electron microscopes. Electronic structures within single atoms can even be probed in real-space[8–10].

Even after the advent of aberration correction technology, magnetic objective lenses with minimal lens aberration coefficients remain indispensable for atomic-resolution TEMs/STEMs. One critical drawback of the current magnetic condenser-objective lens systems for atomic-resolution TEMs/STEMs is that the samples have to be inserted into very high magnetic fields, up to 2~3 T, to realize the short focus length condition essential for atomic-resolution imaging. Figure 1a, b show a schematic cross section of a conventional magnetic objective lens and the magnetic field distribution it produces along the optical axis (z-direction) within the sample region, respectively. In this configuration, samples are inserted between the upper and lower polepieces, which are surrounded by a coil of copper wires. When an electrical current passes through the coil, a magnetic field is created within the bore of the polepieces, and this field is used to focus the incoming high-energy electron beam. Figure 1b also schematically plots the z-component of the magnetic field distribution across the upper and lower polepieces. The maximum field strength is found near the sample position. This high magnetic field can severely hamper atomic-resolution imaging of many important soft/hard magnetic materials such as silicon steel because the material's magnetic and sometimes physical structures can be largely altered, even destroyed, by the strong magnetic field. Thus, Lorentz imaging mode[11], in which the magnetic objective lens is completely switched off or exchanged for so-called Lorentz type lenses[12–14], has been used to observe magnetic materials in TEMs/STEMs at the expense of high spatial resolution. There have been several attempts to develop magnetic field-free objective lens systems for high spatial resolution TEM/STEM imaging, but atomic-resolution imaging under magnetic field-free conditions is still extremely challenging even with the aid of the state-of-the-art aberration correctors and ultrahigh voltage electron sources (see Supplementary Table 1 for a summary of previously reported attainable spatial resolution by TEMs/STEMs in a magnetic field-free environment).

In the present study, we develop a new magnetic objective lens system combined with a state-of-the-art aberration corrector and thereby simultaneously realize atomic resolution electron microscopy (with sub-Å spatial resolution) and a magnetic field-free sample environment. We first confirm the environment around

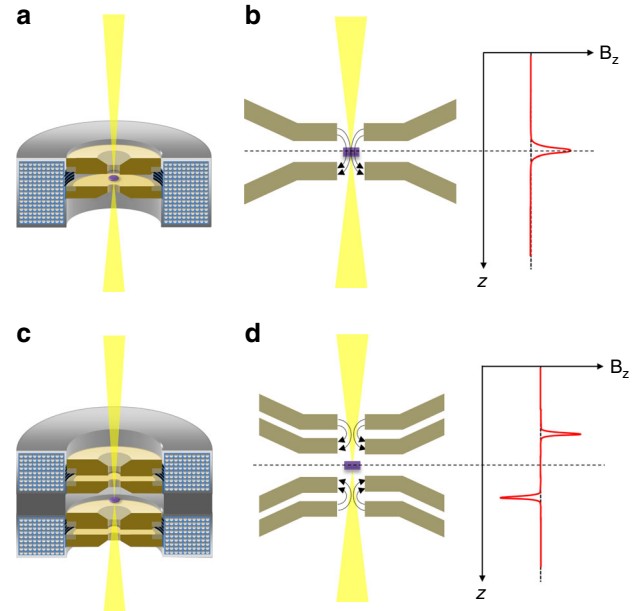

**Fig. 1** Schematic illustration of magnetic objective lens systems for atomic-resolution TEMs/STEMs. **a** Schematic cross section of a conventional magnetic objective lens. Samples (shown as a purple disk) are placed between the upper and lower polepieces (in gold color). **b** Schematic illustration showing the z-component magnetic field ($B_z$) distribution across the upper and lower polepieces when we excite the magnetic objective lens system. The maximum $B_z$ is found near the sample position. **c** Schematic cross section of the new magnetic field-free objective lens system developed in this study. This lens system is composed of two round lenses, and samples are placed in between the FOL and the BOL lenses. The polepieces and coils of the FOL and BOL are exactly in the mirror symmetric configuration with respect to the sample plane, but the polarities of their excitations are opposite, resulting in an anti-symmetric magnetic field distribution across the sample plane. **d** Schematic illustration shows the $B_z$ distribution across this new objective lens. The $B_z$ can be canceled out at the sample position, while the strong magnetic field needed for forming an atomic size electron probe can be located as close as possible to the sample plane

the sample position is magnetic field free through the direct measurement of the three-dimensional magnetic field distribution within the magnetic objective lens, and then observe nonmagnetic crystals to prove through annular dark-field (ADF) STEM imaging that sub-Å spatial resolution is achieved. Finally, we directly observe atomic structures of a typical soft magnetic material (silicon steel) to demonstrate the capability of this new magnetic objective lens system for characterizing atomic structures of magnetic materials.

## Results

**Development of a new magnetic field-free objective lens system.** Figure 1c, d show a schematic cross section of the newly designed objective lens system in this study and the magnetic field distribution it produces along the z-direction within the sample region, respectively. This lens system is basically composed of two round lenses[15]. The front objective lens (FOL) is located in front of the specimen and the back objective lens (BOL) is located behind the specimen, with respect to the incoming electron beam. The magnetic polepieces and coils of the FOL and BOL are in an exact mirror symmetric configuration with respect to the sample plane. However, the polarities of their excitations are opposite, resulting in an anti-symmetric magnetic field distribution across the sample plane. Thus, the z-component of the magnetic fields of

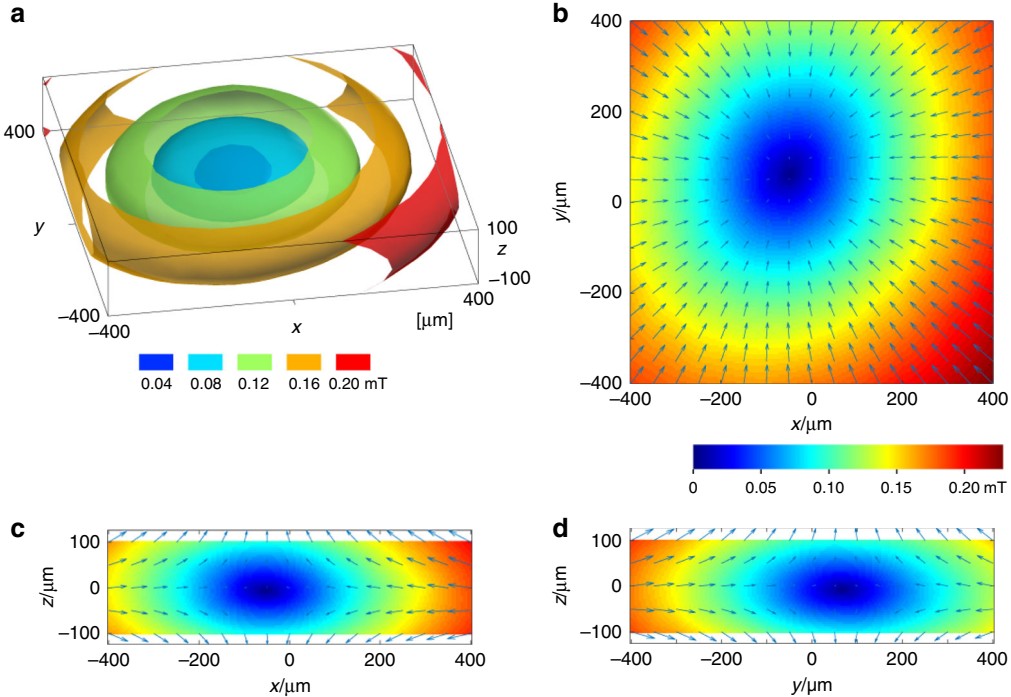

**Fig. 2** Experimentally measured three-dimensional (3D) magnetic field distribution in between the FOL and BOL. **a** 3D magnetic field strength map (shown in color scale) within a $800 \times 800 \times 200\ \mu m^3$ volume between the FOL and BOL. It can be seen that, around the center of the volume, the magnetic field strength is <0.2 mT. **b–d** Two-dimensional (2D) magnetic field vector maps on **b** $xy$ ($z = 0$), **c** $xz$ ($y = 0$), and **d** $yz$ ($x = 0$) planes. The magnetic field vectors point inwards from the outer region to the center of the volume, and diverge to the $\pm z$-direction at the center region

the FOL and BOL can be ideally canceled out at the sample plane. In addition, the radial component of the magnetic field is minimal near the optic axis because of the symmetry of the round lenses. This new lens system should, in principle, realize extremely small residual magnetic fields at the sample position, while placing the strongly excited FOL and BOL close enough to the sample to realize the short focus length condition indispensable for atomic-resolution imaging. Note that the present magnetic objective lens should also be applicable to the aberration-corrected TEM configuration due to its symmetric structure about the sample plane. This magnetic objective lens thus has huge potential for application to other existing TEM/STEM imaging and analytical techniques, such as high-resolution TEM, conventional TEM, electron holography, Lorentz TEM, differential phase contrast STEM imaging, ptychography, electron energy loss spectroscopy and energy dispersive X-ray spectroscopy.

To test this new objective lens concept in practice, we developed such an objective lens system and installed it into a STEM (200 kV accelerating voltage) equipped with a state-of-the-art higher-order aberration corrector, which can correct lens aberrations up to 5th order sixfold astigmatism[16]. Figure 2 shows the experimental three-dimensional magnetic field distribution inside the objective lens directly measured through Hall devices placed on a TEM specimen holder. Figure 2a shows the three-dimensional magnetic field strength map within a $800 \times 800 \times 200\ \mu m^3$ volume in between the FOL and BOL. Since atomic-resolution TEM/STEM observation is normally carried out for very small sample regions, typically within 100 nm in the $x$ and $y$ direction and with a thickness of <50 nm or so, the above volume well covers the region of interest of typical samples for atomic-resolution TEM/STEM observation. Figure 2b–d show two-dimensional magnetic field vector maps on $xy$ ($z = 0$), $xz$ ($y = 0$), and $yz$ ($x = 0$) planes, respectively, with the measured local magnetic field modulus at each position shown in color scale. It is

seen that, while the FOL and BOL are strongly excited, the residual magnetic fields near the sample center become much <0.2 mT, >10,000 times smaller than the value in a conventional magnetic objective lenses used for atomic resolution TEM/STEM imaging. The aberration corrector was also excited during the magnetic field measurement. Thus, we conclude that the stray magnetic fields from the FOL, BOL, and all the other possible sources are well compensated and that an essentially magnetic field-free environment is realized at the sample position.

**Atomic-resolution ADF STEM imaging of a non-magnetic crystal.** Atomic-resolution ADF STEM observations of a GaN crystal were performed using the new objective lens system combined with the higher-order aberration corrector. The aberration corrector was situated between the electron gun and the objective lens so as to correct the lens aberrations of the probe-forming FOL. The lens parameters of the BOL alone (the FOL should have the same value by the symmetry) were experimentally estimated at the accelerating voltage of 200 kV as follows: Cs = 16.9 mm, Cc = 3.45 mm and a focal length of 3.14 mm. The details of the evaluation can be found in the Methods section. The experimental Ronchigram, which is the shadow image of the probe, is shown in Supplementary Fig. 1, obtained with an amorphous carbon film on the sample position. It is seen that the flat phase region extends up to 26 mrad in semiangle. Using a 200 kV accelerated electron beam with a probe-forming aperture of 20 mrad in semiangle, we observed a GaN single crystal along the [111] direction by ADF STEM as shown in Fig. 3a. In the ADF image, only the atomic columns of Ga are visible because of the strong atomic number-dependent contrast. Figure 3b shows the experimental repeat-unit averaged image and the crystal structure model of GaN from the same viewing direction. The Ga-Ga dumbbells of 0.92 Å separation are clearly resolved in real-space. The Fourier transform of the ADF image shown in Supplementary Fig. 2 also confirms sub-Å spatial resolution.

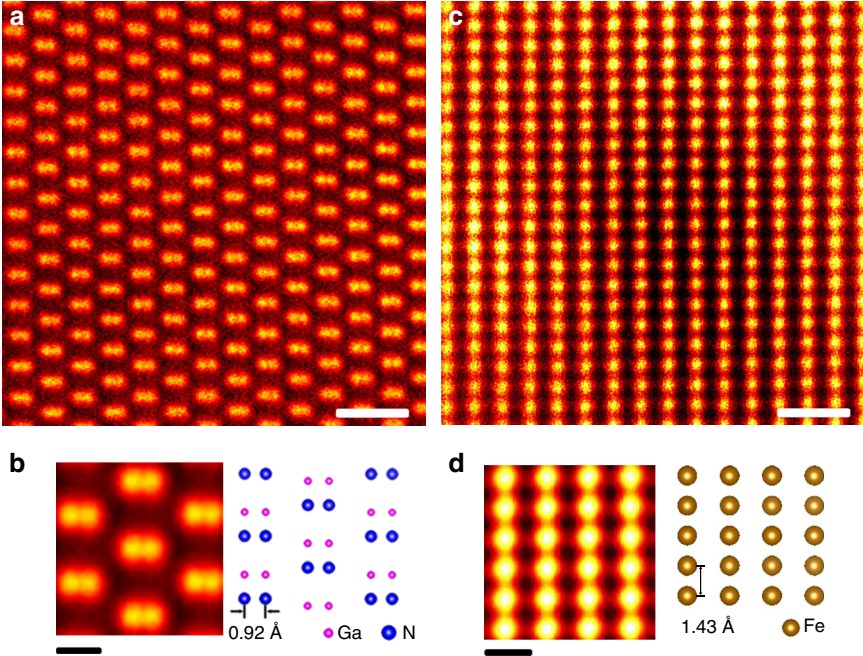

**Fig. 3** Atomic-resolution ADF STEM images of non-magnetic and magnetic crystals in a magnetic field-free environment. **a** Experimental ADF STEM image of a GaN single crystal observed along the [111] direction. Scale bar indicates 0.5 nm. This image is cropped from the averaged image of very fast scan STEM images acquired in ten sequential frames, each containing 512 × 512 pixels at a dwell time of 5 µs/pixel followed by image filtering with the radial difference filter[18], released by the HREM Research, in order to remove background noise. **b** Experimental repeat-unit averaged image from the ADF STEM image and the crystal structure model of GaN projected along the [111] direction. Scale bar indicates 0.2 nm. The crystal structure model was drawn using the VESTA software[19]. Ga-Ga dumbbells of 0.92 Å separation are well resolved. **c** Experimental ADF STEM image of Fe-3%Si observed along the [110] direction. Scale bar indicates 0.5 nm. This image is cropped from the averaged image of very fast scan STEM images acquired in ten sequential frames, each containing 512 × 512 pixels at a dwell time of 4 µs/pixel. **d** Experimental repeat-unit averaged image from the ADF STEM image and the crystal structure model of Fe-3%Si projected along the [110] direction. Scale bar indicates 0.2 nm

## Atomic-resolution ADF STEM imaging of a magnetic crystal.

We also observed the atomic structure of a grain-oriented silicon steel sheet (Fe-3wt%Si) as shown in Fig. 3c. Figure 3d shows the experimental repeat-unit averaged image and the crystal structure model of Fe-3wt%Si from the same viewing direction. Grain-oriented silicon steel is one of the most important engineering soft magnetic materials and atomic-resolution characterization of individual defects has long been sought. However, it has hitherto been extremely challenging to directly observe their atomic structures by TEMs/STEMs since samples are largely deformed or even destroyed by the strong lens magnetic field, and moreover the sample magnetic field makes it extremely difficult to reliably tune the lens astigmatism well enough to allow atomic-resolution observations. However, as shown in Fig. 3c, using our new objective lens system we can directly and clearly resolve the atomic structure of the silicon steel along the [110] direction (see also Supplementary Fig. 3 for an atomic-resolution image of a grain boundary in the silicon steel). Thus, direct atom-resolved imaging in a magnetic field-free environment is finally realized in electron microscopy, enabling unprecedented atomic-level structural characterization of magnetic materials.

In summary, atomic-resolution electron microscopy in a magnetic field-free environment is finally realized using a newly developed objective lens system combined with state-of-the-art aberration correction technology. This imaging capability should allow atomic structure characterization of any magnetic material, which has hitherto been extremely challenging in TEMs/STEMs. Since this objective lens can also be applied to other existing TEM/STEM imaging and analytical techniques, the present achievement should have considerable far-reaching effects across the entire field of transmission electron microscopy.

## Methods

**STEM sample preparation**. GaN single crystal and grain-oriented Fe-3%Si alloy were mechanically polished with diamond suspension, and then thinned by Ar ion-beam milling to obtain electron transparency.

**Magnetic field distribution inside the new objective lens system**. To map the three-dimensional (3D) magnetic field distribution inside the new magnetic objective lens system, we developed a sample holder with a small Hall device placed on the specimen position, which can measure the 3D magnetic field distribution at each sample position. We then systematically moved the sample positions in between the FOL and BOL and measured the 3D magnetic fields at each point within the range $x: \pm 400 \, \mu m$, $y: \pm 400 \, \mu m$, $z: \pm 100 \, \mu m$. The step size between sample positions was 50 µm. The magnetic fields at 1445 points along three $x, y, z$ directions were measured. The isosurfaces and the color maps were obtained by interpolating the measured values. In addition, we experimentally measured that the maximum magnetic field change on the sample position during the sample insertion into the objective lens, without turning off the objective lens, did not exceed 0.8 mT.

**Aberration coefficients of the new objective lens system**. To experimentally estimate the lens parameters of the new magnetic field-free objective lens system, we first assumed the lens parameters of the FOL and BOL should be the same because of their structural symmetry. Then, we estimated the lens parameters of the BOL in TEM mode.

To estimate the focal length of the BOL, we used the following equation,

$$f = \frac{M}{(M+1)^2} L \tag{1}$$

where $f$ is the focal length, $M$ is the magnification of the BOL and $L$ is the distance between the sample and image planes. For a sample, we used MAGICAL (Norrox Scientific Ltd.), a standard sample for TEM magnification calibration. Here, we physically measured the distance between the sample and the image plane (selected area (SA) aperture plane) and used an already-known size SA aperture to measure the magnification. The focal length was estimated to be 3.14 mm.

To estimate the Cs of the BOL, we observed an Au nanoparticle sample under Scherzer focus condition and obtained the phase contrast transfer function (PCTF) by taking the Fourier transform of the image. Using the first zero of the PCTF, we

defined the point resolution to be $\delta = 0.473$ nm. Then, we estimated the Cs value of the BOL to be 16.9 mm.

To estimate the Cc (following the caution discussed in ref. [17]) of the BOL, we changed the energy of incident electron beam and measured the defocus change systematically. The defocus change $\Delta f$ due to the energy shift of the incident beam $\Delta E$ is given by

$$\Delta f = \mathrm{Cc}\frac{\Delta E}{E} \qquad (2)$$

Using the experimental linear relationship between the defocus change ($\Delta f$) and the energy shift ($\Delta E$) of incident electron beam as shown in Supplementary Fig. 4, we estimated the Cc value to be 3.45 mm.

These measured lens parameters are compared with another existing Lorentz type lens as shown in Supplementary Table 2.

**Microscope set-up**. The base electron microscope for this experiment is a commercially available cold field emission type 200 kV STEM (ARM-200CF, JEOL Ltd.). We exchanged the normal objective lens system to the newly developed magnetic field-free type objective lens system. In this system, standard double-tilt sample holders are acceptable. The tilt angle ranges are $x$-axis: $\pm 10°$ and $y$-axis: $\pm 10°$.

**ADF STEM imaging conditions**. ADF STEM images shown in this study were obtained with the following optical conditions. The probe-forming aperture semiangle was set to 20 mrad for the GaN case and 24 mrad for the Fe-3% Si case. The inner and outer angles of the ADF detector were estimated to be about 39 mrad and 100 mrad, respectively.

## Data availability
Data are available from the corresponding author upon reasonable request.

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

## Acknowledgements
We thank A. Kimura and N. Saito for their assistance in TEM specimen preparation. This work was supported by the JST SENTAN Grant Number JPMJSN14A1, Japan. A part of this work was supported by the JSPS KAKENHI Grant numbers JP17H01316 and JP17H06094. A part of this work was conducted in Research Hub for Advanced Nano Characterization, The University of Tokyo, under the support of "Nanotechnology Platform" (project No.12024046) by MEXT, Japan. This research was partly supported under the Discovery Projects funding scheme of the Australian Research Council (Project No. DP160102338).

## Author contributions
N.S. and Y.K. designed the study and N.S. wrote the paper. Y.K. developed the magnetic field-free objective lens and measured magnetic field inside the objective lens. Y.K., S.M., A.N., H.S., A.K., N.S. performed the STEM experiments, and T.S. performed the data analysis. T.M., S.D.F., and Y.I. contributed to the discussion and comments. N.S. directed the entire study.

## Additional information

**Competing interests:** The authors declare no competing interests.

