## [Peer Review File · Nature Communications]

Editorial Note: This manuscript has been previously reviewed at another journal that is not operating a transparent peer review scheme. This document only contains reviewer comments and rebuttal letters for versions considered at Nature Communications .

Reviewers' comments:

Reviewer #1 (Remarks to the Author):

The authors have carefully revised their manuscript. My suggestions (Reviewer 1) have all been fully considered. Overall the response to the reviewers is very carefully written and the arguments are carefully set and worked out.

I repeat my assessment of the text that I gave in the first evaluation. (1) High-resolution electron microscopy, i.e. microscopy with atomic resolution, has so far not been possible for magnetic materials. This was a painful limitation for research. (2) Electron microscopy works with lenses consisting of electromagnetic fields generated by current-carrying ring-shaped coils. These magnetic fields, which are required for high-resolution imaging, are very strong. Therefore, it has not been possible to study magnetizable or magnetic materials at high resolution in an electron microscope. The idea of the authors to modify the Riecke-Ruska lens, which is used practically everywhere and throughout today's electron microscopes, so that an area is created within the objective lens that is practically free of magnetic field, is ingenious. (3) No one has come up with this idea before, which would have been obvious, because the Riecke-Ruska lens is symmetrically composed of two partial lenses. As the authors describe it, one simply has to polarize the current direction of the two partial lenses in the opposite direction. (4) The authors show in their work that (a) as proven by field measurements, the magnetic field strength at the location of the sample is very small (below 0.3 mTesla; in practice negligible). And they already show atomic resolution in the first version of the manuscript. Furthermore, the work is written very carefully and prudently. This work will be much noted, and it will undoubtedly enable the application of high-resolution electron microscopy in the field of magnetic materials, now on a large scale. This definitely justifies the publication in the journal. (5) In the revised second version, a sensational image of the atomic structure of iron-silicon steel material has now been inserted, which of course significantly improves the quality of the manuscript. Fe-Si is a standard magnetic steel. It is a great achievement that this can now be studied at high resolution; nobody would have dared to think of this before! This is the first atomic resolution image of the atomic structure of such a difficult magnetic material in a high-resolution electron microscope ever!

To summarize: It is my intense recommendation to accept this manuscript for publication [redacted]. The content is innovative as far as the completely new technique is concerned, and sensational as far as the atomic imaging of a magnetic material is concerned. This paper will be highly appreciated by the microscopy community and it will be the starting point of many related applications.

PS: I have read the comments of the other reviewers. Apparently they don't have much knowledge in the field of electron microscopy. The Lorentz microscopy, which one of the reviewers cites, is usually performed with the objective lens turned off. Thus, the magnification is too low by a factor of about 1000 to obtain atomic resolution. In addition, these reviewers do not seem to have noticed (this is very puzzling) that this manuscript is not reporting on a technique to measure the magnetization in the sample. The manuscript reports on the first high quality atomic image of a magnetic sample ever. This is a sensational result. To prevent the publication in a journal by demanding additional results in fields that are not important for the publication and are therefore not addressed at all, cannot be the task of referencing such an excellent work.

Reviewer #2 (Remarks to the Author):

“Atomic resolution electron microscopy in a magnetic field free environment” by Shibata et al

I appreciate the authors' efforts to revise the manuscript based on my comments and resubmit it to Nature Communication. Although the authors have addressed some of my concerns and criticisms two problems remain (the first one is significant!).

1) [redacted]

2) I am not saying the development of the field-free polepiece on high resolution imaging is not significant. However, the performance of the instrument, say both Table 1 and Table S1, in the reply is very misleading. For instance, in Table S1, why the spatial resolution of JEOL non-aberration corrected Lorentz microscope was not included? Around 2000-2005 JEOL has made commercial Lorentz microscopes (i.e., the one at Tohoku University). At that time, the achievable resolution was 4Å (I only found the attached document showing 5Å resolution in TEM, JEOL News, Vol 42, No. 1, page 2-7 (2007)). Please note, the polepiece has a gap of 8mm, thus it is not a high-resolution polepiece but aims at large sample tilt holder and inserting coils for magnetization. With a small gap the resolution of the polepiece can be further improved.

Reviewer #3 (Remarks to the Author):

I appreciate the efforts the authors have made to try and change their paper, however I remain of the opinion that this paper would be best suited to the journal Ultramicroscopy as I stated in my original review. The authors certainly demonstrate a significant development in imaging of atomic structure in a field free environment. As such I believe that paper could have been published with likely minor changes in Ultramicroscopy. However the real novelty would be using this resolution to image magnetic structure on that scale, which is not shown here, only physical structure. I understand the difficulties which will be involved in imaging magnetic structure at atomic

resolution, this is actually going to be quite a large jump in terms of theory and channeling effects etc to consider. The paper title does not specify magnetic imaging but I expect most reader would be expecting that from the system described, certainly for this journal.

So whilst the advance to image atomic scale in a field free environment is impressive I really do not think this alone justifies publication in this journal. If true magnetic imaging was demonstrated at this scale that would be a different matter of course.

Response to the Reviewer #1

We thank the reviewer for reviewing our revised manuscript again. We are gratified by the reviewer's very strong recommendation of this manuscript for publication [redacted]. In particular, we are grateful for the reviewer's enthusiasm on our observation of the atomic structure of Fe-Si steel. As the reviewer comments, our achievement will be the first high quality atomic image of such a very difficult magnetic sample in TEM ever, and could be a sensational result for many researchers in the material science and device engineering community. We thank the reviewer again for their strong support of our achievements.

Response to the Reviewer #2

We thank the reviewer for reviewing our revised manuscript again. We are happy to hear that most of the concerns and criticisms are already addressed. The reviewer still raised two remaining technical concerns. We carefully checked the reviewer's comments. The summary of our responses to the points raised by the reviewer and the corresponding changes made to the manuscript are italicised and interspersed between the reviewer's report below.

[redacted]

[redacted]

[redacted]

Please note, the polepiece has a gap of 8mm, thus it is not a high-resolution polepiece but aims at large sample tilt holder and inserting coils for magnetization. With a small gap the resolution of the polepiece can be further improved.

We do not consider Table 1 to be misleading. In our opinion, it shows a fair comparison between the two lenses by highlighting the most important lens parameters. It makes clear that these lens parameters are significantly improved by our new design, especially the much shorter focal length for large demagnification that is essential to obtain a stable, noise-tolerant, atomic-size electron probe for STEM. We reiterate that this comparison was introduced in response to the reviewer's previous suggestion that combining that previous Lorentz lens with aberration-correction would enable atomic-resolution imaging. We believe this comparison shows this to be very unlikely.

*To the best of our knowledge and despite an additional intense literature survey, we know of no reports on this kind of Lorentz lens with a smaller gap size. The following schematic shows a cross sectional view of the Lorentz lens system later used by Schofield et al. as reported in D. Shindo et al., Scr. Mater., **48**, 851-856 (2003).*

Fig. Cross section of the polepiece with a single gap.

In this Lorentz lens system, the specimen is not inserted inside the polepiece gap but rather above the upper polepiece. In this configuration, lens parameters are mainly determined by the physical distance between the sample plane and the center plane of the polepiece gap. Therefore, we expect there should be no significant improvements in the lens parameters by just changing the polepiece gap width. However, being reluctant

to speculate on the performance of a lens system which has never actually been made, we have instead inserted the following sentence in the revised manuscript.

It should be noted that this previous Lorentz type lens is not designed for ultra-high resolution imaging.

Response to the Reviewer #3

We thank the reviewer for reviewing our revised manuscript again. Imaging magnetic structure on the atomic scale via transmission electron microscopy would be a phenomenal result. This is obviously not that paper. Nevertheless, we strongly believe that the present achievements possess great novelty. Atomic resolution imaging of any kind of magnetic material, which we explicitly demonstrate through the example of Fe-Si steel in the revised manuscript, is of huge potential impact. Our work overcomes the long-standing challenge in electron microscopy of the strong magnetic field from objective lens hindering observing atomic structure of soft/hard magnetic materials: in the words of reviewer #1 “this is the first atomic resolution image of the atomic structure of such a difficult magnetic material in a high-resolution electron microscope ever!” We believe this will have tremendous impact not only amongst electron microscopists, but also amongst materials scientists and device engineers using electron microscopy. Especially since, as the reviewer acknowledges, the title does not specify magnetic imaging, we do not share the reviewer’s expectation that readers will expect this paper to demonstrate magnetic imaging at atomic resolution, and as such have not sought to further address this point. In previous revisions, we added to further clarity to the body text and conclusions on the resolution at which magnetic imaging is shown, as distinct from the imaging of atomic structure (atom locations) within magnetic materials. We believe this should sufficiently forestall any confusion to readers about this point. As stressed in our previous response and revised manuscript, we strongly believe our achievement will have substantial scientific impact across a wide range of electron microscopy, materials science and engineering fields. Atomic-resolution magnetic field imaging, when it is finally achieved, will indeed be significant. But it is just one of many possible applications our present achievements will help enable in future.

Reviewers' comments:

Reviewer #2 (Remarks to the Author):

[redacted]